# Optimizing direct RT-LAMP to detect transmissible SARS-CoV-2 from primary nasopharyngeal swab samples

Dawn M. Dudley[1], Christina M. Newman[1], Andrea M. Weiler[2], Mitchell D. Ramuta[1], Cecilia G. Shortreed[1], Anna S. Heffron[1], Molly A. Accola[3], William M. Rehrauer[3], Thomas C. Friedrich[4], David H. O'Connor[1] *

1 Department of Pathology and Laboratory Medicine, University of Wisconsin-Madison, Madison, Wisconsin, United States of America, 2 Wisconsin National Primate Research Center, University of Wisconsin-Madison, Madison, Wisconsin, United States of America, 3 University of Wisconsin Hospitals and Clinics, Madison, Wisconsin, United States of America, 4 Department of Pathobiological Sciences, University of Wisconsin-Madison, Madison, Wisconsin, United States of America

* dhoconno@wisc.edu

## Abstract

SARS-CoV-2 testing is crucial to controlling the spread of this virus, yet shortages of nucleic acid extraction supplies and other key reagents have hindered the response to COVID-19 in the US. Several groups have described loop-mediated isothermal amplification (LAMP) assays for SARS-CoV-2, including testing directly from nasopharyngeal swabs and eliminating the need for reagents in short supply. Frequent surveillance of individuals attending work or school is currently unavailable to most people but will likely be necessary to reduce the ~50% of transmission that occurs when individuals are nonsymptomatic. Here we describe a fluorescence-based RT-LAMP test using direct nasopharyngeal swab samples and show consistent detection in clinically confirmed primary samples with a limit of detection (LOD) of ~625 copies/µl, approximately 100-fold lower sensitivity than qRT-PCR. While less sensitive than extraction-based molecular methods, RT-LAMP without RNA extraction is fast and inexpensive. Here we also demonstrate that adding a lysis buffer directly into the RT-LAMP reaction improves the sensitivity of some samples by approximately 10-fold. Furthermore, purified RNA in this assay achieves a similar LOD to qRT-PCR. These results indicate that high-throughput RT-LAMP testing could augment qRT-PCR in SARS-CoV-2 surveillance programs, especially while the availability of qRT-PCR testing and RNA extraction reagents is constrained.

## Introduction

There are more than 13.8 million reported severe acute respiratory syndrome coronavirus 2 (SARS-CoV-2) infections in the United States as of December 3rd, 2020 (https://www.cdc.gov/coronavirus/2019-ncov/cases-updates/cases-in-us.html). The actual number of infections is likely far greater since testing remains limited. Asymptomatic individuals contain similar levels

**Data Availability Statement:** All relevant data are within the manuscript and its Supporting information files. Additional details about protocols

and results are posted at https://openresearch.labkey.com/Coven/wiki-page.view?name=lamp-testing.

**Funding:** This work was supported in part by the Office of Research Infrastructure Programs/OD under grant P51OD011106 awarded to the Wisconsin National Primate Research Center at the University of Wisconsin-Madison (awarded to DHO and TCF) and more information is available at https://orip.nih.gov/. This work was also supported by the Rapid Acceleration of Diagnostics (RADX) program through the National Institutes of Health (grant number 144 AAI2136 awarded to DHO and TCF) and more information is available at https://www.nih.gov/research-training/medical-research-initiatives/radx. Lastly, this work was also funded by the Wisconsin Alumni Research Foundation COVID19 Accelerator Challenge (grant number 135 AAH9333 awarded to DHO and TCF) and more information is available at https://www.warf.org/programs-events/community/covid-19-updates/. There was no additional external funding received for this study. The funders had no role in study design, data collection and analysis, decision to publish, or preparation of the manuscript.

**Competing interests:** The authors have declared that no competing interests exist.

of SARS-CoV-2 in the upper respiratory tract as symptomatic individuals [1–6]. Furthermore, 17 out of 24 (71%) presymptomatic patients had positive viral cultures 1 to 6 days before the onset of symptoms [1]. Symptom-based testing is not sufficient for controlling SARS-CoV-2 transmission and emphasizes the need for expanded nucleic acid surveillance of asymptomatic/presymptomatic individuals.

Conventional SARS-CoV-2 testing relies on RT-PCR amplification of virus-specific nucleic acids extracted from nasopharyngeal (NP) swabs. However, shortages of nucleic acid extraction and RT-PCR reagents as well as RT-PCR instrumentation, remain a problem [7]. In addition, this test is expensive ($25/sample for reagents) and turn-around time is often several days [8–10]. Alternative nucleic acid extraction methods and "direct" testing that does not require nucleic acid extraction are important to expand testing while reducing time and cost. For example, the SalivaDirect method recently approved under an FDA EUA utilizes saliva without RNA extraction into an RT-PCR assay [11].

Reverse transcription loop-mediated isothermal amplification (RT-LAMP) has been for point-of-need diagnostic testing for several pathogens, including SARS-CoV-2 [12–21]. RT-LAMP assays are an alternative method for rapidly detecting the presence of specific nucleic acids in samples, with colorimetric or fluorescent visualization of results. RT-LAMP assays are inexpensive (~$7/sample), high-throughput (can be run in a 96-well format), do not necessarily require nucleic acid purification, and give rapid results (~60–90 minutes from set-up to results). Previously published manuscripts demonstrate proof-of-principle for SARS-CoV-2 testing by RT-LAMP using either contrived samples with free nucleic acid or extracted RNA from primary samples [12–14,16–22]. Minimally processed primary NP swab samples are more challenging, since biological inhibitors such as nucleases may hinder amplification or degrade RNA. Here we describe the LOD of direct RT-LAMP without RNA isolation on primary NP swab samples.

Direct RT-LAMP is an example of a lower sensitivity, fast-turnaround test that requires minimal equipment that can be used in a point-of-need setting. SARS-CoV-2 antigen tests provide a similar point-of-need, lower sensitivity, and rapid test. The CDC recently released guidance highlighting the importance of point-of-need antigen testing for screening asymptomatic individuals without known SARS-CoV-2 exposure [23]. These tests are also in limited supply. The importance of quick-turnaround and inexpensive tests is becoming increasingly recognized to help mitigate transmission of SARS-CoV-2 [24–26]. The current gold-standard qRT-PCR test has proven to have a turn-around time of several days and therefore does little to move highly contagious individuals into isolation before they transmit the virus [25]. Analytical modeling of different screening strategies shows that very frequent, inexpensive, and even poorly sensitive testing is predicted to sufficiently isolate positive individuals and prevent widespread transmission better than low frequency highly sensitive testing [27]. Direct RT-LAMP, therefore, has the potential to become an important addition to the currently available SARS-CoV-2 testing arsenal.

In this study, we focused on characterizing and optimizing direct RT-LAMP without RNA isolation and with primary NP swab samples with known SARS-CoV-2 status. We demonstrate the limit of detection (LOD) of direct swab RT-LAMP in primary swab samples as well as modifications that help improve sensitivity but don't rely on the same materials required for traditional qRT-PCR methods. We characterized the use of Lucigen QuickExtract (QE) lysis buffer, guanidine hydrochloride addition, an alternative RNA isolation method, and several primer sets and combinations targeting different gene regions. Systemic evaluation of these modifications with primary samples will be useful to other groups designing RT-LAMP workflows for SARS-CoV-2 surveillance.

## Materials and methods

### Sample collection

Residual, completely de-identified NP swab samples were provided by the University of Wisconsin-Madison Hospitals and Clinics (UWHC) and the Wisconsin State Laboratory of Hygiene (WSLH) under biosafety protocol B00000117 (IRB 2016–0605), and their use was not considered human subjects research by the University of Wisconsin-Madison School of Medicine and Public Health's Institutional Review Board. Samples were not specifically collected for this study. Samples were collected into a variety of transport media including universal transport media (UTM), viral transport media (VTM), and phosphate-buffered saline (PBS), stored at 4˚C for up to 7 days, and transported to the laboratory at room temperature. Upon arrival at the laboratory, samples were stored at either 4˚C (for immediate same-day use) or -80˚C until use in RT-PCR or RT-LAMP assays.

### qRT-PCR

Viral load analysis was performed after samples arrived in our laboratory. RNA was isolated using the Viral Total Nucleic Acid kit for the Maxwell RSC instrument (Promega, Madison, WI) following the manufacturer's instructions. Viral load quantification was performed using a qRT-PCR assay developed by the CDC to detect SARS-CoV-2 (specifically the N1 assay) and commercially available from IDT (IDT cat # 10006770) (Coralville, IA) [28]. The assay was run on a LightCycler 96 or LC480 instrument (Roche, Indianapolis, IN) using the Taqman Fast Virus 1-step Master Mix enzyme (Thermo Fisher, Waltham, MA). The LOD of this assay is estimated to be 200 genome equivalents/ml in swab media. To determine the viral load, samples were interpolated onto a standard curve consisting of serial 10-fold dilutions of *in vitro* transcribed SARS-CoV-2 N gene RNA kindly provided by Nathan Grubaugh (Yale University).

### RT-LAMP

The experiments we describe here were modified from the SARS-CoV-2 RT-LAMP assay developed by Zhang et al. [13]. We used fluorescent-based detection with Warmstart LAMP reagents and the included fluorescent dye (New England Biolabs, NEB cat # E1700L). We tested primer sets developed in previous studies targeting several SARS-CoV-2 genes as shown in S1 Table [13,14,22,29–33]. Of note, the Color-Orf1a primers and Lamb-Orf1a primers are identical but were used at different concentrations per the protocols developed by each originating lab. The final 1X primer concentrations are listed in S1 Table. For each reaction, a 10X stock of all 6 primers was combined with Warmstart mastermix and water in 25μl reactions following the manufacturer's recommendations. Unless otherwise stated, 1μl RNA transcript of the SARS-CoV-2 N-gene obtained by Dr. Nathan Grubaugh (S2 Table), 1μl of synthetic SARS-CoV-2 RNA transcript (Twist Biosciences; RNA control 2), 1μl of gamma-irradiated SARS-CoV-2 (BEI; NR-52287; isolate USA-WA1/2020), or 1μl primary NP swab sample were tested in each RT-LAMP reaction. Unless otherwise stated, all serial dilutions were performed in water. For reactions testing guanidine hydrochloride addition to the RT-LAMP mastermix, a final concentration of 40mM stock was used in the mastermix. Except where otherwise specified, samples were run on a Roche Lightcycler 96 instrument (Roche Diagnostics) using an 80-cycle program with the SYBR Green channel at 65˚C (495–497 nm absorption; 517–520 nm emission) with each cycle representing data collected every 30 seconds. The presented Cq value represents the cycle number where detection of the RT-LAMP amplicon begins. The more template in the reaction and the more efficient the reaction conditions are, the earlier

the detection begins. For experiments determining the appropriate volume of direct swab sample addition for highest RT-LAMP efficiency, a 60-cycle program with data collection every 20 seconds was used.

The specificity of the primers tested in this manuscript were established in previous publications [13,14,22,29,31–33]. Briefly, the Gene-N-A primers, used in most of the experiments in this manuscript, did not cross-react with SARS-CoV-1 N-gene by RT-LAMP and were not anticipated to cross-react with other common respiratory pathogens based on sequence comparisons (communication with Nathan Tanner). The Gene-N2, Gene-E1, and As1e primers were aligned against other coronaviruses and respiratory pathogens and the only pathogen with >80% identity to SARS-CoV-2 in more than one of the 6 primers required for RT-LAMP was SARS-CoV-1 (communication with Nathan Tanner and Brad Langhorst), which hasn't circulated in humans since 2004. The Color Genomics primers were tested for cross-reactivity against 51 organisms for EUA approval and were not cross-reactive to any organism tested [33].

## Sample lysis

A subset of samples were treated with LucigenQE RNA Extraction Solution (Lucigen, Middleton, WI) in a 1:1 ratio as described in Ladha et al. [34]. Briefly, NP swab media was combined with an equivalent volume of LucigenQE and briefly vortexed. Samples were then incubated for 5 minutes at 95°C, cooled on ice, and maintained until the addition to the RT-LAMP reaction.

## Statistical analysis

To assess improvement in quantification cycle (Cq) values using sample lysis with LucigenQE or RNA isolation, mean Cq values were calculated for each sample. Mean Cq values were not normally distributed for either dataset so we used a nonparametric equivalent to a paired t-test, the Wilcoxon signed rank test with continuity correction, for each set of paired samples.

To examine whether sample vRNA load and/or treatment were significantly associated with a positive RT-LAMP result, we used logistic regression with the RT-LAMP result as the dependent variable. For our analysis, an equivocal result in which one replicate was positive while the other was negative, was conservatively treated as negative. We coded RT-LAMP results for each sample tested by each method as a dichotomous outcome with positive samples coded as "1" and negative or equivocal samples coded as "0". For explanatory variables, we chose qRT-PCR vRNA load, with samples greater than $10^3$ copies/μl coded as "1" and samples less than $10^3$ copies/μl coded as "0", and group, designated as either 5μl RNA, 1μl lysed, or 1μl direct addition.

All statistical analyses were performed in RStudio (v. 1.2.1335) using R (v. 3.6.0) [35].

## Results

### Limit of detection with RNA transcript and Gene-N-A primers

To determine a limit of detection for the RT-LAMP assay, serial 10-fold dilutions of RNA transcripts containing the N-gene were tested in RT-LAMP reactions with Gene-N-A primers in duplicate in 3 independent assays. RNA transcripts were diluted in RNase-free water. Consistent detection of RNA was achieved when $1\times10^3$ copies or greater of RNA/μl was added into the reactions (Fig 1A). To obtain a more precise LOD, the transcript was diluted 1:2 starting at $5\times10^3$ copies/μl down to 78 copies/μl and each concentration was run in 10 replicates. Nine of ten replicates at 625 copies/μl were positive, while 6/10 were positive at 312 copies/μl and 5/10

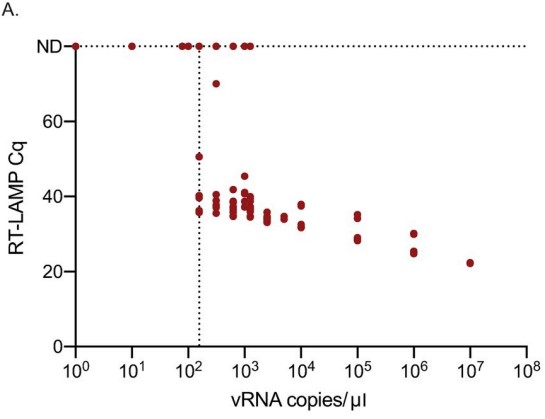

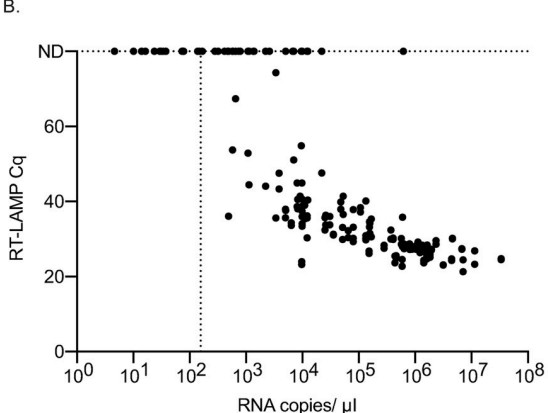

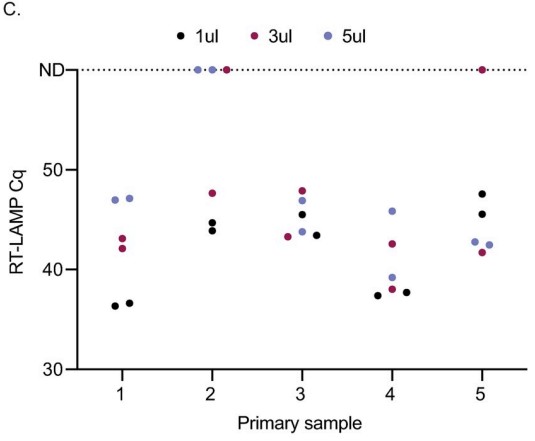

**Fig 1. Detection of SARS-CoV-2 by RT-LAMP from transcript or primary NP swab samples.** (A) The quantification cycle (Cq) relative to each transcript copy number is plotted. Samples that were not detectable were plotted on the line labeled ND for all graphs at Cq of 60 or 80, equal to the total number of cycles. The vertical line is set at the lowest dilution where positive samples were detected using the transcript input for all graphs (156 vRNA copies/μl). Each replicate is plotted on all graphs. (B) Detection of 106 SARS-CoV-2 positive primary NP swab samples relative to their in-house viral load value. (C) RT-LAMP Cq of five SARS-CoV-2 positive primary NP swab samples with different swab input volumes.

were positive at 156 copies/µl (Fig 1A). Thus, we can consistently detect 625 copies of input into the reaction but can detect down to 156 copies of input in half of the reactions. Zero reactions were detected as positive at 100 copies/µl or below.

## Limit of detection with primary nasopharyngeal swab samples

Residual NP swab samples from 106 patients with diagnosed SARS-CoV-2 and 31 negative samples were tested directly by RT-LAMP in duplicate. Additionally, RNA was isolated from the positive samples and tested by qRT-PCR with a transcript standard for quantitation. A total of 63/106 (59%) samples tested positive by RT-LAMP and 106/106 by qRT-PCR (S3 Table). Another 13 samples were equivocal by RT-LAMP, with one of two replicates positive. As shown in Fig 1B, the LOD of primary samples was similar to that seen with the RNA transcript. For the rest of the analysis, we focus on the ability of this assay to detect samples with a viral load of $> 1 \times 10^3$ copies/µl as a conservative LOD based on our transcript LOD of 625 copies/µl. 63/78 (81%) samples with viral RNA copy numbers greater than $1 \times 10^3$ copies/µl were detected by RT-LAMP, whereas 0/28 samples with concentrations $< 1 \times 10^3$ copies/µl were detected positive and 3/28 were equivocal. All 31 samples that tested negative for SARS-CoV-2 by clinical laboratories also tested negative by RT-LAMP (S3 Table samples 107–137). Negative samples were not tested by our internal qRT-PCR assay. Five of the 31 samples that tested negative for SARS-CoV-2 tested positive for other respiratory pathogens including influenza A, rhinovirus, respiratory syncytial virus, influenza B, and human metapneumovirus. In this limited sample set, the Gene-N-A primers showed 100% specificity for SARS-CoV-2.

## RT-LAMP is inhibited by adding larger volumes of primary sample

To determine whether adding larger volumes of primary NP swab samples could improve the sensitivity of the assay, 1, 3, and 5µl of swab samples from 5 primary SARS-CoV-2-positive samples were tested side-by-side. All replicates were detected as positive when 1µl was added directly into the RT-LAMP reaction (Fig 1C). However, one of the two replicates from two samples tested negative when 3µl of the sample was added and both replicates from one sample tested negative when 5µl of the sample was added into the RT-LAMP reaction. Furthermore, Cq thresholds were higher with the addition of higher volumes of sample. Therefore, we chose to use 1µl of straight swab sample in subsequent experiments.

## Lysis buffer improves the sensitivity of RT-LAMP

To determine whether treatment with lysis buffer improves the sensitivity of RT-LAMP to detect SARS-CoV-2 in clinical samples without the use of traditional nucleic acid isolation methods, we treated 72 clinical samples with a range of SARS-CoV-2 vRNA loads in a 1:1 ratio with LucigenQE as described by Ladha et al. [34]. We then compared the fluorescent RT-LAMP Cq values between 1µl of lysed sample and 1µl of the same samples added directly. The addition of 1µl of NP swab eluate directly into the RT-LAMP reaction resulted in positive detection in both replicates of 46/72 (64%) known SARS-CoV-2-positive samples and in 1 of 2 replicates in 6 additional samples (Fig 2A, S3 Table). Treatment of the same 72 samples with LucigenQE resulted in detection of both replicates for 56/72 (78%) samples, an additional 10 samples that were undetectable before. An additional 4 samples that were negative when tested directly were equivocal when treated with LucigenQE. Focusing on samples above $1 \times 10^3$ copies/µl, 45/53 (85%) samples were positive without extraction while 53/53 (100%) were positive with LucigenQE extraction. Without RNA extraction an additional 5 samples were equivocal indicating that extraction resulted in more consistent detection of positive samples in both replicates above the $1 \times 10^3$ copies/µl threshold. While 0/19 samples with viral loads below

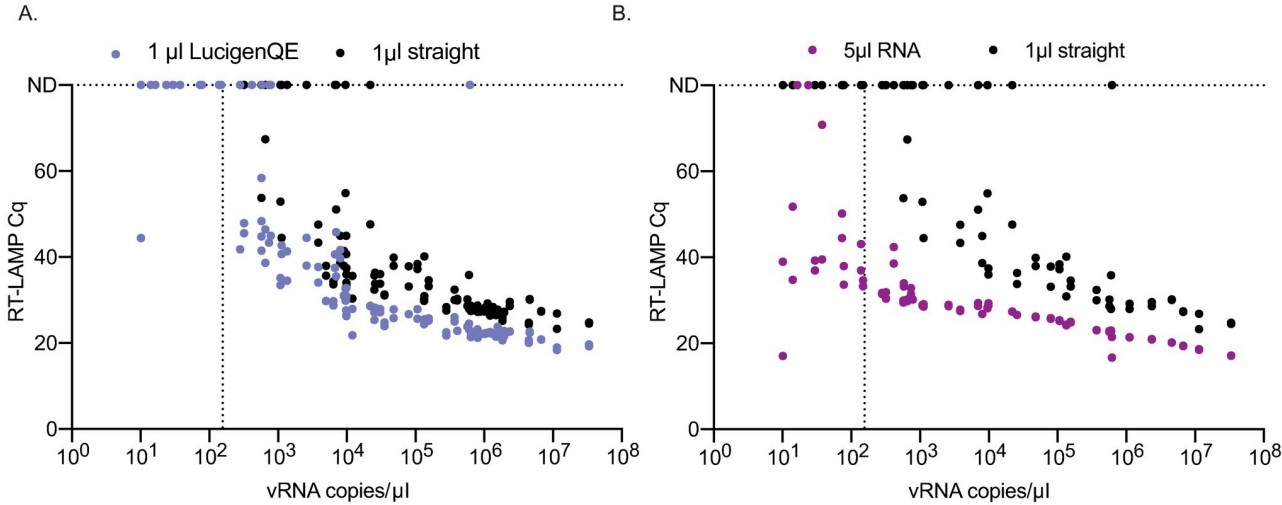

**Fig 2. Comparison of the detection of SARS-CoV-2 positive primary NP swab samples after using direct sample addition to either LucigenQE treatment or isolated vRNA.** (A) Comparison of Cq values between samples treated with or without LucigenQE and run by RT-LAMP with 1µl of sample. (B) Comparison of Cq values between RT-LAMP assays run with 1µl straight swab or 5µl of isolated and purified vRNA. vRNA copies/µl are reported for the starting sample concentration before RNA purification and concentration.

$1x10^3$ copies/µl were positive with 1µl straight, 4 of 19 samples were detectable after lysis in both replicates and 4 additional samples were detected in 1 of 2 replicates. Mean Cq for LucigenQE-treated samples were significantly lower (Cq = 38.62) than those for directly added samples (Cq = 49.08) (Wilcoxon signed rank test, V = 1830, p = $1.67E^{-11}$), suggesting that lysis treatment improves the efficiency of amplification in the LAMP reaction (Fig 2A). We also examined whether sample vRNA load and/or treatment were significantly associated with RT-LAMP detection results. We found that direct addition of 1µl of untreated NP swab was associated with a decreased odds of detecting a positive RT-LAMP result (OR = 0.20, 95% CI = 0.04–0.65, p = 0.015) while the most important factor associated with a positive result was a sample vRNA load of greater than $1x10^3$ copies/µl (OR = 88.5, 95%CI = 25.74–434.87, p = $1.88E^{-10}$).

## RNA isolation improves the limit of detection of RT-LAMP to levels similar to qRT-PCR

One of the primary reasons the direct RT-LAMP assay is less sensitive than the diagnostic qRT-PCR assay is because the qRT-PCR assay uses 5µl of concentrated and purified RNA as input. To determine whether the RT-LAMP assay would perform to a similar level of detection if the same input was used, 5µl of purified RNA was used in RT-LAMP assays from a subset (n = 44) of COVID-19-positive NP swab samples. The starting sample viral loads ranged from $1.01x10^1$ to $1.14x10^7$ copies/µl. 25 of 44 had concentrations of virus above the RT-LAMP LOD of $1x10^3$ vRNA copies/µl. Of the 44 samples tested with 1µl of the direct swab, 18 tested positive (43%) and 7 were equivocal between replicates (Fig 2B and S3 Table). When 5µl of purified RNA was used in the reactions instead, 42 of 44 samples (95%) tested positive (Fig 2B, S3 Table). Note that the process of RNA isolation used in this experiment concentrated the RNA approximately 3-fold from the concentration presented in Fig 2B and that 5µl of that concentrated RNA was used, while 1 µl of the starting sample concentration was used in the 1µl straight reactions. Samples with 100-fold lower vRNA copies/µl were detected after RNA isolation and concentration by RT-LAMP. Adding purified RNA brings the possible LOD of

detection down to 150 copies of input of a 10 copies/μl sample, which was detected in 4 of 6 primary samples tested within this viral load range. This is similar to the detection of 156 copies of transcript as shown in Fig 1A. Of the 25 samples with viral loads over the $1 \times 10^3$ copies/μl threshold, 18 (72%) were positive when adding straight while 25/25 (100%) were positive with RNA isolation.

Overall, the direct addition of 1μl NP swab was associated with reduced odds of a positive RT-LAMP result (OR = 0.0054, 95%CI = 0.00023–0.041, p = $2.56E^{-05}$). Similar to the results for lysis buffer treatment, samples with qRT-PCR vRNA loads greater than $1 \times 10^3$ copies/μl had significantly increased odds of a positive RT-LAMP result (OR = 49.35, 95%CI = 8.22–966.29, p = 0.00044). Lastly, the Cq values were lower for all detected samples when 5μl of RNA was added (mean Cq = 31.42) instead of 1μl straight sample (mean Cq = 58.72), indicating faster and more robust detection with concentrated and purified RNA (Wilcoxon signed rank test, V = 903, p = $1.71E^{-08}$) (Fig 2B).

## Alternative primers improve efficiency

Multiple SARS-CoV-2 RT-LAMP primer sets have been published or included in manuscripts on preprint servers since we began our experiments. We compared the efficiency of several alternative primer sets either alone or in combination with the Gene-N-A primers used in our initial studies (S1 Table). Two primary NP swab samples with high concentrations of SARS-CoV-2 (NP1:$1.09 \times 10^9$ copies/μl, NP2:$4.28 \times 10^7$ copies/μl) as well as gamma-irradiated SARS-CoV-2 (BEI) were used to screen different primer combinations using the same fluorescent RT-LAMP conditions. First, a set of primers previously established [14,30] and used to obtain an FDA EUA by Color Genomics was tested with each primer alone and in different combinations (Fig 3A). Several primers and primer combinations resulted in a lower Cq value across the board than the Gene-N-A gene primers suggesting improved reaction efficiency. The primer combinations including Color-N/Color-E and Color-N/Color-ORF1a yielded the lowest Cq values in all samples.

Next, we compared the Gene-N-A, Color-N, Color-E, and Color-N/E combination to second-generation primers described in Zhang et al. [22] (Gene-N2, Gene-E1 and As1e). We found that As1e, originally published by Rabe et al. [29], yielded the lowest Cq value followed closely by a combination of As1e with the two primers targeting the Gene-N2 and Gene-E1 genes designed by Zhang et al. (Fig 3B). The Color-N primer yielded similar Cq values to the triple combination primer set from Zhang et al. We also tested the Color-N, Color-E1, and Gene-N-A gene primer sets against additional published primers that target ORF1a (Lamb, Yu, El-Tholoth, and Zhang primers) either alone or in combination and compared them to the Gene-N-A gene primer set (S1 Table). The Color-N primer produced the lowest Cq value in this set (S1 Fig).

We then tested As1e, Color-N, As1e/Color-N/Gene-E1 primer set with and without guanidine hydrochloride, as recommended by Zhang et al., with Twist RNA SARS-CoV-2 template [22]. Under these conditions, guanidine hydrochloride improved detection with most primer sets with the exception of the Gene-N-A primer set (Fig 3C). Using the Twist RNA, the primer set that detected samples at the lowest dilutions was the Color-N primer or combination of As1e/Color-N/Gene-E1 with guanidine hydrochloride, though only in one of two replicates. The As1e primer with and without GuHCl as well as the combination As1e/Color-N/Gene-E1 with GuHCl often yielded the lowest Cq value.

Lastly, to determine which primer set or combination worked best with primary NP swab samples, fourteen samples with viral loads ranging from $7.33 \times 10^4$-$1.52 \times 10^8$ vRNA copies/μl were tested using 1μl of straight swab sample with the most promising primer combinations

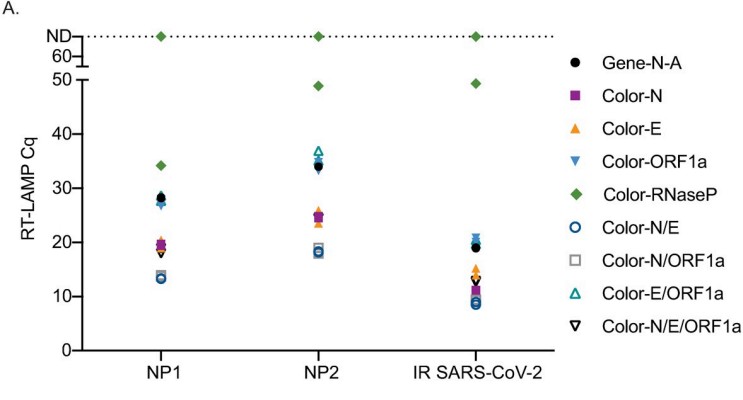

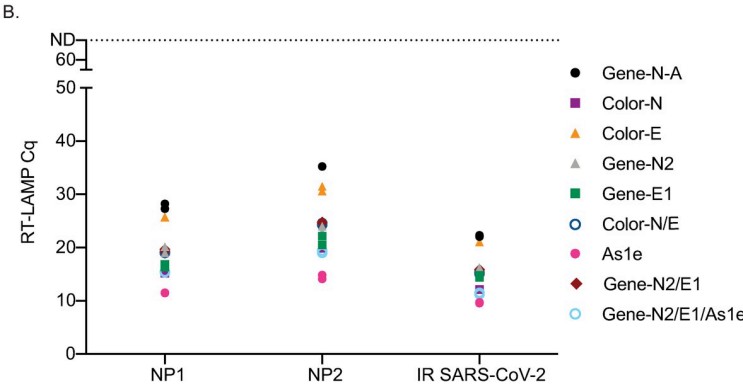

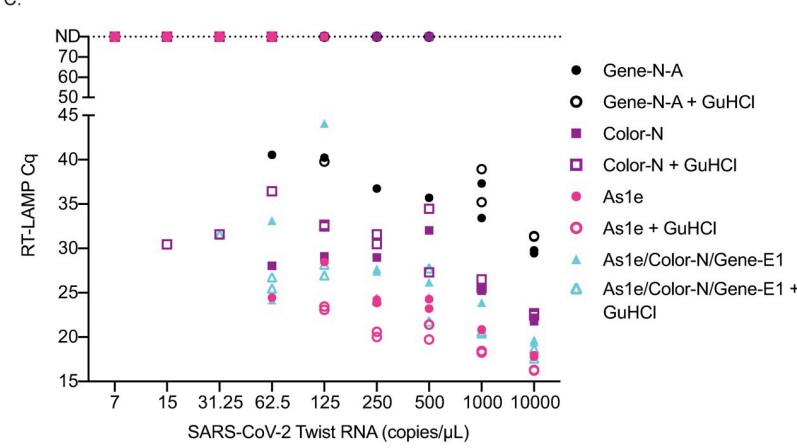

**Fig 3. Comparison of RT-LAMP Cq value on primary NP swab samples, irradiated SARS-CoV-2 or SARS-CoV-2 TWIST RNA amplified with different primer sets.** Reactions were run in duplicate and both replicates are shown on the graphs. Samples that were not detectable are plotted on the ND line at Cq 80. (A) Comparison of Cq values using Color Genomics primers to the Gene-N-A primers on two primary NP swab samples and irradiated SARS-CoV-2. (B) Comparison of Zhang et al. [22] primers to a subset of Color Genomics primers and Gene-N-A primers on two primary samples and irradiated SARS-CoV-2. (C) Comparison of the Cq values obtained when using the best primers and combinations of primers across different dilutions of SARS-CoV-2 TWIST RNA with and without GuHCl.

A.

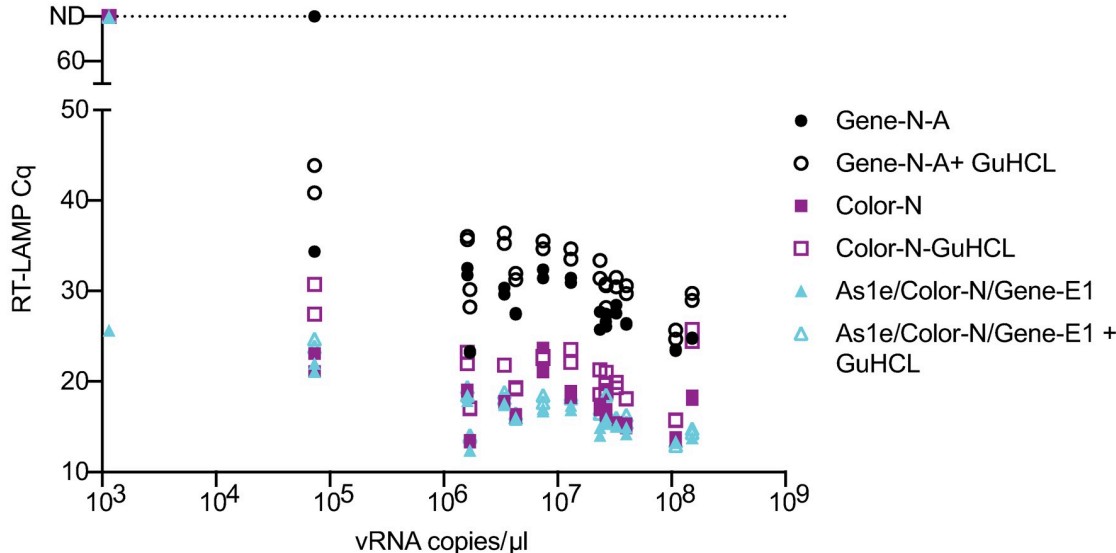

B.

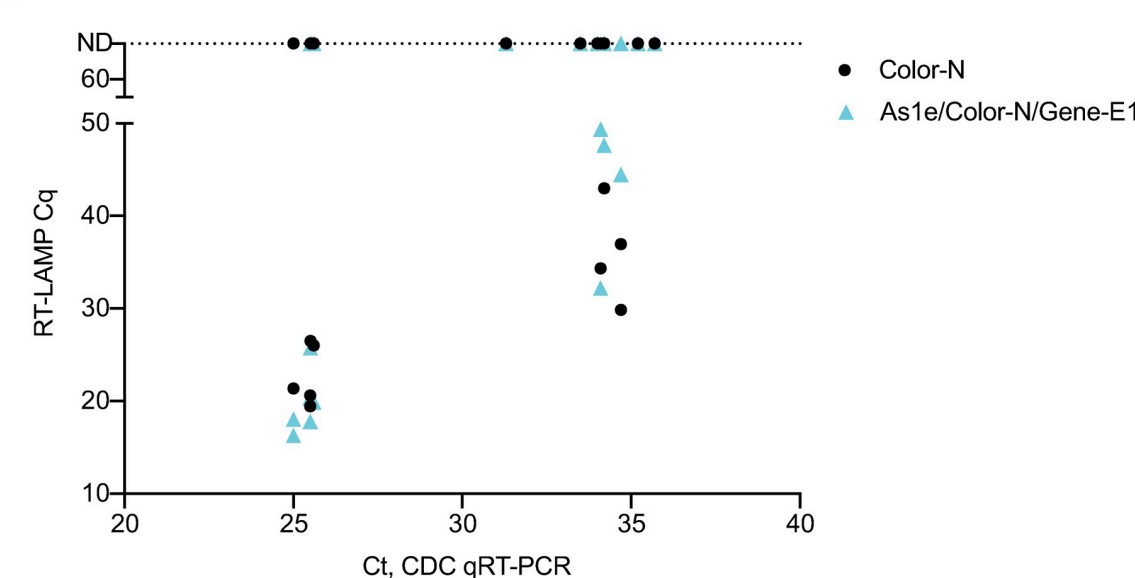

**Fig 4. Comparison of the best-performing primers and combinations on additional primary NP swab samples with varying levels of virus with and without GuHCl.** (A) Comparison of three primer sets or combinations either with or without GuHCl on 14 primary NP swab samples with high viral load copy numbers. (B) Comparison of the Color-N primer set to As1e/Color-N/Gene-E1 on primary NP swab samples with lower levels of virus. These samples were not run by our in-house viral load test and therefore Ct value obtained from the hospital is reported.

with and without GuHCl. Both the Color-N primer set alone and the As1e/Color-N/Gene-E1 combination yielded the lowest Cq value across the different levels of virus (Fig 4A). Guanidine hydrochloride did not improve detection in primary samples as seen with the Twist RNA. Twelve additional primary NP swab samples with high Ct values ranging from 25 to 35 were tested with the Color-N and As1e/Color-N/Gene-E1 combination. Detection of these samples

was very similar between the two primer conditions (Fig 4B). Note that several samples were equivocal at Ct 25 for both primer sets equivalently and that we might expect better detection at that threshold. Since we do not eliminate inhibitors in the samples and since these samples were stored for several days and freeze-thawed between the time the hospital ran the sample to acquire the Ct value and the time the sample was run in our assay, some of the replicates became undetectable. We would expect to more consistently detect virus at a Ct of 25 in both replicates in fresh samples.

## Discussion

Frequent, widespread testing is considered the best mitigation strategy to control SARS-CoV-2 before a vaccine or effective therapy becomes widely available in 2021. Traditional qRT-PCR is sensitive but is time-consuming and reliant on very specific reagents that are in short supply. RT-LAMP has become a promising alternative to qRT-PCR, but the sensitivity of this assay has been poorly characterized from primary NP swab samples when added directly into the reaction. Many published studies establish a LOD based on free RNA transcript or isolated RNA from cultured virus of around 100 copies/rxn. These LODs apply to RT-LAMP only when purified RNA is used as input. They do not apply to direct RT-LAMP methods containing whole virions in primary samples, which also contain host enzymes and other host components. In this study, we established that 625 copies/reaction was necessary in order to detect RNA transcripts consistently in 9/10 reactions in our assay. In primary samples we rarely detected virus in samples with a vRNA load of less than $5 \times 10^2$-$1 \times 10^3$ vRNA copies/μl, establishing this threshold as a conservative LOD. While not as sensitive as methods with RNA extraction first, this LOD range is sufficient for SARS-CoV-2 surveillance to detect virus in individuals with the minimal amount of virus necessary to isolate the virus and therefore most likely to transmit the virus [1,36–38].

The LOD of $1 \times 10^3$ copies/μl is likely due to inefficiencies associated with virus lysis at 65°C during the LAMP reaction and possible degradation of liberated RNA by enzymes, including RNases, present in primary samples. Indeed, adding more sample, including more host enzymes and media with potential inhibitors, reduced detection. On the other hand, RNA extraction eliminates these inhibitors and allows 100% consistent detection above 10 copies/μl (150 copies/reaction). Lysis with LucigenQE was compatible with RT-LAMP and improved our sensitivity to detect SARS-CoV-2 in primary samples with vRNA loads less than $10^3$ copies/μl. When compared with direct addition, we were able to increase our detection of true positives for both replicates from 61% to 78% after lysis treatment. In samples with viral loads above $1 \times 10^3$ copies/μl we improved detection to 100%. Guanidine hydrochloride has also been shown to improve sensitivity in other studies and while our results showed better detection with synthetic Twist RNA, bringing the LOD down to 62.5 copies/μl, the same improvement was not observed in primary samples in our hands. The largest increase in sensitivity occurred with RNA isolation prior to RT-LAMP using an alternative RNA isolation method to those approved for the CDC qRT-PCR assay. With RNA isolation we detected 95% of the samples detected by qRT-PCR, including those with the lowest viral loads.

There are now many primers available targeting different regions of SARS-CoV-2. In this work, we tested several sets to iteratively choose which primer set worked best with primary NP swab samples. We found that several primer sets performed more efficiently than the Gene-N-A primers we used in most of our experiments. The two best performing primer sets that were nearly indistinguishable in performance with both high and low viral load primary samples were a combination of As1e/Color-N/Gene E1 and the Color-N primer set alone.

For this study, we chose to use fluorescent RT-LAMP for detection rather than colorimetric detection. Using fluorescence enabled analysis of the differences in the Cq values providing a quantitative evaluation of how each condition changed the efficiency of the assay. However, when considering how to deploy RT-LAMP in the field, our group has developed a mobile RT-LAMP workflow that uses saliva and colorimetric readouts for low cost and portability [39]. Additional work needs to be done to determine whether the benefits of fluorescent detection can be inexpensively migrated to decentralized point-of-need testing.

Many studies are comparing RT-LAMP to RT-PCR results presented as Ct value, rather than vRNA copies/µl. We chose to focus on comparing methods and determining LODs based on a standard qRT-PCR assay performed in our laboratory with a quantitative standard, rather than Ct values generated by the varying sources of our primary samples. Diagnostic laboratories have transitioned between different methods as reagents were available and as new assays became available, which means that the Ct values can vary. By comparing all our results to the vRNA copies/ml that we generated in our lab using a consistent primer set and protocol (CDC qRT-PCR assay) that targeted the same gene (and primers) as our SARS-CoV-2 RT-LAMP assay, we were able to ensure our comparisons were consistent across all samples.

Overall, we have shown that direct RT-LAMP using fluorescent detection can detect SARS-CoV-2 in primary NP swab samples with viral loads greater than $1\times10^3$ vRNA copies/µl. We were able to improve this slightly with the quick and low-cost addition of Lucigen lysis buffer to the reaction and could detect 100% of samples above the $1\times10^3$ copies/µl threshold. We also saw improvement in efficiency with several alternative primer sets. While direct RT-LAMP is not as sensitive as qRT-PCR, it is sufficient to detect the levels of virus that are necessary to culture virus from a sample. This means that this assay detects people who are likely to transmit the virus for a significant reduction in cost, time, and reagents relative to qRT-PCR. Direct RT-LAMP costs approximately $7 per sample while qRT-PCR costs $25. RT-LAMP from set-up to results can be performed in 60–90 minutes depending on the scale of the assay (more samples take longer to set up on the front end, but the reaction time is 30 minutes), while qRT-PCR requires 60 minutes for RNA extraction, 30 minutes to set-up qRT-PCR, and an additional 90 minutes for the qRT-PCR run to finish. One proposed way to utilize this test could be to surveil large numbers of individuals who are then directed toward diagnostic testing by qRT-PCR if they test positive by this test, significantly reducing the burden on diagnostic labs and their resources. This test, if transitioned to a colorimetric version, could be set up at point-of-need sites with minimal equipment and could provide same-day test results to individuals in a work or school environment. For this test to be used in a clinical or diagnostic capacity, additional clinical validation would be required. Overall, direct RT-LAMP is an important addition to the repertoire of currently available tests to identify samples containing SARS-CoV-2 nucleic acids.

## Supporting information

**S1 Fig. Comparison of Gene-N-A, Color-N, and Color-E primers to several ORF1a -targeting primer sets and combinations with two primary samples and irradiated SARS-CoV-2.** Samples that were not detectable were plotted on the ND line set at Cq 80, the hißghest cycle number in our assay.
(TIF)

**S1 Table. Primer sequences and final 1X concentrations used in the RT-LAMP reactions.**
(DOCX)

**S2 Table. Nucleotide sequence of the RNA transcript provided by Dr. Nathan Grubaugh.**
(DOCX)

**S3 Table. qRT-PCR N-gene viral loads and RT-LAMP Cq values from 106 primary NP swab samples run in duplicate with 1ul of swab sample or a subset of NP swab samples run in duplicate with either 1μl of primary samples treated with Lucigen QuickExtract or 5μl of extracted vRNA.**
(DOCX)

## Acknowledgments

We thank Nathan Tanner for discussions about primers and optimizations of the RT-LAMP assay. We thank Brad Langhorst for specificity data for NEB primer sets. We thank Dr. Nathan Grubaugh for the Sars-CoV-2 N-gene transcript. The following reagent was deposited by the Centers for Disease Control and Prevention and obtained through BEI Resources, NIAID, NIH: SARS-Related Coronavirus 2, Isolate USA-WA1/2020, Gamma-Irradiated, NR-52287.

## Author Contributions

**Conceptualization:** Dawn M. Dudley, Christina M. Newman, Cecilia G. Shortreed, Thomas C. Friedrich, David H. O'Connor.

**Data curation:** Dawn M. Dudley, Christina M. Newman, Andrea M. Weiler, Mitchell D. Ramuta, Cecilia G. Shortreed, Molly A. Accola.

**Formal analysis:** Dawn M. Dudley, Christina M. Newman, Andrea M. Weiler, William M. Rehrauer.

**Funding acquisition:** Thomas C. Friedrich, David H. O'Connor.

**Investigation:** Dawn M. Dudley, Christina M. Newman, Andrea M. Weiler, Mitchell D. Ramuta, Cecilia G. Shortreed, David H. O'Connor.

**Methodology:** Dawn M. Dudley, Christina M. Newman, Cecilia G. Shortreed, William M. Rehrauer, Thomas C. Friedrich.

**Project administration:** Anna S. Heffron.

**Resources:** Andrea M. Weiler, Anna S. Heffron, Molly A. Accola, William M. Rehrauer, David H. O'Connor.

**Supervision:** Dawn M. Dudley, Thomas C. Friedrich.

**Validation:** Dawn M. Dudley, Christina M. Newman, Andrea M. Weiler, Cecilia G. Shortreed, David H. O'Connor.

**Visualization:** Dawn M. Dudley, Christina M. Newman, Mitchell D. Ramuta.

**Writing – original draft:** Dawn M. Dudley, Mitchell D. Ramuta.

**Writing – review & editing:** Dawn M. Dudley, Christina M. Newman, Andrea M. Weiler, Mitchell D. Ramuta, Cecilia G. Shortreed, Anna S. Heffron, Molly A. Accola, William M. Rehrauer, Thomas C. Friedrich, David H. O'Connor.

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
