## [Decision Letter · Decision Letter 0]

26 Oct 2020

PONE-D-20-29372

Optimizing direct RT-LAMP to detect transmissible SARS-CoV-2 from primary nasopharyngeal swab and saliva samples

PLOS ONE

Dear Dr. Dudley,

Thank you for submitting your manuscript to PLOS ONE. After careful consideration, we feel that it has merit but does not fully meet PLOS ONE’s publication criteria as it currently stands. Therefore, we invite you to submit a revised version of the manuscript that addresses the points raised during the review process.

Authors are asked to prepare the manuscript in accordance with the comments of the reviewers and respond to their comments.

We look forward to receiving your revised manuscript.

Kind regards,

Ruslan Kalendar, PhD

Academic Editor

PLOS ONE

Journal Requirements:

2. Please clarify if the biological samples used in your study were:

(1) from an established biobank (if so please provide the name and a link)

(2) specifically collected for this study or not

(3) whether the samples were collected through a medically prescribed test

(4) whether the samples were completely de-identified before researchers accessed the samples.

3. Please provide the catalog number for the:

a) coronavirus qRT-PCR assay used in this study and

b) 4 Warmstart LAMP reagents and the included fluorescent dye (New England Biolabs, NEB).

4. Please provide the sequence of the  in vitro transcribed SARS-CoV-2 N gene RNA provided by Nathan Grubaugh.

Reviewers' comments:

Reviewer's Responses to Questions

**Comments to the Author**

1. Is the manuscript technically sound, and do the data support the conclusions?

Reviewer #1: Yes

Reviewer #2: No

2. Has the statistical analysis been performed appropriately and rigorously? 

Reviewer #1: Yes

Reviewer #2: Yes

3. Have the authors made all data underlying the findings in their manuscript fully available?

Reviewer #1: Yes

Reviewer #2: Yes

4. Is the manuscript presented in an intelligible fashion and written in standard English?

Reviewer #1: Yes

Reviewer #2: Yes

5. Review Comments to the Author

Reviewer #1: Thanks to the authors for this very careful work on a diagnostic system for SARS-Cov-2. Given the context of the current pandemic and recurring shortages of reagents, any serious studies evaluating the performance of diagnostic tests are important.

I have some comments on the body of the text and a general remark on the introduction and discussion of this work.

It is obvious that the system presented (without conventional prior DNA extraction) lacks sensitivity unlike the gold standards. The authors from the introduction endeavor to cite all the literature which justifies that beyond a certain Ct, the virus is no longer cultivable. This notion has already been known for a long time and is true for almost all viruses: the fewer viruses there are the more difficult it is to cultivate it ... However, I understand that this notion is important to promote this LAMP type system but I think you shouldn't be so exclusive and avoid extrapolating. Is there any literature linking Ct and infection in "real life" and not in vitro? This kind of study is obviously complicated to implement that's why your introduction and conclusion must be moderate. If a person in your family was at 30Ct, would you find it normal that she thinks herself negative with the behavior that this causes, moreover some patients may have been badly sampled. In short, you justified that your system is good at detecting contagious people despite your low sensitivity based on in vitro studies. I think it deserves to be qualified a little.

Finally here are some minor comments.

The CDC System reference is missing.

You should include earlier that the heating is used for inactivation of biological samples in the field.

Line 212 there is a problem with the text font size

Your percentages are oddly expressed. You should rather focus on the samples> 106 which you are supposed to detect (63/77, 83% 46/53, 87% 18/25, 72%)

Figure 4, how do you explain that samples at 25CT are not detected?

The paragraph on saliva for me is not of interest, if you think that the conditions of transport and storage are not comparable it is useless to develop this part further. You should simply delete it.

Reviewer #2: The manuscript entitled “Optimizing direct RT-LAMP to detect transmissible SARS-CoV-2 from primary nasopharyngeal swab and saliva samples” by Dudley and colleagues describes the analytical sensitivity of previously published RT-LAMP assays for SARS-CoV-2. While the manuscript is well prepared, the study has little novelty and several major concerns:

Major concerns:

1. The author’s interpretation in relation to the poor sensitivity of the assays (1x10^5-1x10^6 copies/ml) they have evaluated is not fully justified. Although in vitro studies showed no or low rates of recovery of replication competent virus in cell culture from specimens with weaker RT-qPCR CT values or lower viral loads, the purpose of COVID-19 testing is not only to determine the transmissibility of the virus. The analytical sensitivity of COVID-19 tests is critical to identify pre-symptomatic cases, to identify patients with severe disease who may have cleared the virus from upper respiratory samples but is still sick from COVID-19 and also for epidemiological purposes. With such a poor sensitivity of their assays it is likely that a large number of tests will end up with false negative results with adverse outcomes.

2. The authors claim to apply their assay as POC is not very encouraging considering the fact that switching to colorimetric detection methods will further reduce the sensitivity of the test.

3. The specificity of any of the assays were not addressed. It is not clear what is the rate of false positive results from these assays.

4. No proper clinical validation was done. So, the clinical performance characteristics of the assays in comparison with standard RT-qPCR assays are not clear.

5. It is somewhat surprising that the LOD for pure transcripts (Fig 1A) is two log higher than extracted RNA from samples (Fig 2B).

6. Different primer sets were tested in a much smaller set of samples compared to Gene-N-A primers.

7. Colorimetric saliva results were not correctly presented. The results need to be blindly called and then compared against NP swab RT-qPCR results.

8. There are now hundreds of RT-qPCR and RT-LAMP methods published including many extraction free PCR with much better analytical and clinical sensitivity. The authors need to discuss how their tests and evaluations provide additional benefits over existing assays.

6. PLOS authors have the option to publish the peer review history of their article (what does this mean?). If published, this will include your full peer review and any attached files.

Reviewer #1: No

Reviewer #2: No

---

## [Author Response · Author response to Decision Letter 0]

16 Dec 2020

Response to Reviewers

PLOS ONE Editor comments

*Please note that all line numbers correspond to the line numbers in the unmarked manuscript.

1. Please ensure that your manuscript meets PLOS ONE's style requirements 

The documents have been updated to conform to the style templates provided. 

2. Please clarify if the biological samples used in your study were:

(1) from an established biobank (if so please provide the name and a link)

(2) specifically collected for this study or not

(3) whether the samples were collected through a medically prescribed test

(4) whether the samples were completely de-identified before researchers accessed the samples.

The Sample Collection section was updated to include the requested information (page 5, lines 93-98). 

3. Please provide the catalog number for the:

a) coronavirus qRT-PCR assay used in this study and

b) 4 Warmstart LAMP reagents and the included fluorescent dye (New England Biolabs, NEB).

These catalogue numbers have been included in the RT-LAMP section of the Materials and Methods (page 5, line 108 and page 6, line 117).

4. Please provide the sequence of the in vitro transcribed SARS-CoV-2 N gene RNA provided by Nathan Grubaugh.

The sequence for the in vitro transcribed SARS-CoV-2 N gene RNA is now included as S2 Table.

5. We note that you have included the phrase “data not shown” in your manuscript. Unfortunately, this does not meet our data sharing requirements. PLOS does not permit references to inaccessible data. We require that authors provide all relevant data within the paper, Supporting Information files, or in an acceptable, public repository. 

We included “data not shown” only for samples that tested negative by our assay that were also negative by diagnostic testing because there is nothing to show on a graph. We now include these negative samples on our table of samples and now refer to that data in S3 Table where the negative nature of the results from these samples is shown (page 10 line 199).

Reviewer comments:

Reviewer #1: 

1. It is obvious that the system presented (without conventional prior DNA extraction) lacks sensitivity unlike the gold standards. The authors from the introduction endeavor to cite all the literature which justifies that beyond a certain Ct, the virus is no longer cultivable. This notion has already been known for a long time and is true for almost all viruses: the fewer viruses there are the more difficult it is to cultivate it ... However, I understand that this notion is important to promote this LAMP type system but I think you shouldn't be so exclusive and avoid extrapolating. Is there any literature linking Ct and infection in "real life" and not in vitro? This kind of study is obviously complicated to implement that's why your introduction and conclusion must be moderate. If a person in your family was at 30Ct, would you find it normal that she thinks herself negative with the behavior that this causes, moreover some patients may have been badly sampled. In short, you justified that your system is good at detecting contagious people despite your low sensitivity based on in vitro studies. I think it deserves to be qualified a little.

The authors agree with the sentiment of this reviewer. In order to address these concerns, the authors wish to de-emphasize identifying contagious individuals. It remains unknown what viral load threshold is required for onward transmission of SARS-CoV-2. While we don’t think anyone disagrees that identifying people with the highest viral loads is likely identifying those who are most contagious, we would like to instead emphasize the utility of this method to implement frequent surveillance for SARS-CoV-2. This assay is not as sensitive as qRT-PCR, but it is faster, cheaper, and more accessible allowing access to asymptomatic individuals who could be frequently tested. qRT-PCR, which cannot keep up with the demand for testing of symptomatic individuals and return results quickly, remains the gold standard but is itself insufficient for frequent surveillance testing.

The sensitivity of LAMP as described in this manuscript rivals other quick tests, such as good antigen tests, that are also in limited supply. There is growing evidence that less sensitive tests that can be administered multiple times per week may be more beneficial than a very sensitive test that provides results more slowly and is administered inconsistently. 

The authors have updated the manuscript (changes to the abstract on page 2 lines 27-29 and 32-33, a paragraph removed in the introduction (page 3, paragraph between lines 49-50) and a new paragraph added (page 4, lines 69-81) into the introduction) to emphasize the value of a test like this to supplement diagnostic testing. 

2. The CDC System reference is missing.

We have included the catalogue number for the components associated with the qRT-PCR assay that was used to generate viral loads in this manuscript (IDT, 10006770) (page 5, line 108). The protocol follows that of the CDC and we have included the reference to that protocol as well on page 5, line 108. 

3. You should include earlier that the heating is used for inactivation of biological samples in the field.

Since heating was only used for the saliva-based system and it was suggested that this be removed from the manuscript, all references to heating the samples were removed.

4. Line 212 there is a problem with the text font size

The font size was fixed at old line 212 (new line 217).

5. Your percentages are oddly expressed. You should rather focus on the samples> 106 which you are supposed to detect (63/77, 83% 46/53, 87% 18/25, 72%)

We thank the reviewers for this suggestion, we agree that this would be helpful to emphasize throughout our comparisons. We did identify an error in that there were 78 samples with viral loads above the 1x10^3 copies/�l threshold from the primary samples in Figure 1B. We apologize for the error and have fixed this in the manuscript. To address this comment, we added a description about why we will focus on samples with viral loads greater than our threshold of 1x10^3 copies/ul to be more conservative than the transcript estimate of 625 copies/ul on pages 9-10, lines 193-195. We then incorporated the percentages of samples that were positive above our 1x10^3 copies/ul limit of detection for each condition we tested and highlighted the improvements these conditions made to the detection of samples that fell at or above this limit of detection. This can be found in lines 224-226, 264-266, and 360-361. 

6. Figure 4, how do you explain that samples at 25CT are not detected?

The ability to detect primary swab samples that have not been purified, but are added directly into the RT-LAMP reaction, is dependent not only on the concentration of the virus in the sample, but also the inhibitors in the sample. Figure 1 shows that there are many undetectable samples with viral loads above the level we believe we can detect with purified RNA. This is also very evident in Figure 2B. When we isolate RNA and purify it away from inhibitors of the primary sample, we detect everything above the 156 copies/ul threshold. 

The variability in nucleases and the media the samples were collected in in the primary sample means that different thresholds of virus may be detectable in different samples. Specifically related to Figure 4, this is the only dataset we were unable to acquire in-house viral loads representative of the viral load of the sample at the time of our assay. Storage for days at 4C and a freeze/thaw of the samples occurred between the time the Ct value was acquired at the hospital and the time that we ran the samples in this assay. Therefore, several samples were equivocal at a Ct of ~25 (all samples were positive in at least 1 replicate), which is likely due to a combination of inhibitors in the samples and the handling of these particular samples between the assay providing the Ct value and our assay. For all other datasets in the manuscript we obtained in-house viral loads after the same storage conditions as our RT-LAMP assay to compare our data for quantitative purposes. This caveat about the data presented in Figure 4 has been included in the text on pages 15 and 16, lines 322-327 and further discussion of the role of inhibitors are included on page 17, lines 352-357.

7. The paragraph on saliva for me is not of interest, if you think that the conditions of transport and storage are not comparable it is useless to develop this part further. You should simply delete it.

The authors have removed the reference to the saliva-based version of this test throughout the manuscript and will describe this in greater detail elsewhere (1).

Reviewer #2: 

1. The author’s interpretation in relation to the poor sensitivity of the assays (1x10^5-1x10^6 copies/ml) they have evaluated is not fully justified. Although in vitro studies showed no or low rates of recovery of replication competent virus in cell culture from specimens with weaker RT-qPCR CT values or lower viral loads, the purpose of COVID-19 testing is not only to determine the transmissibility of the virus. The analytical sensitivity of COVID-19 tests is critical to identify pre-symptomatic cases, to identify patients with severe disease who may have cleared the virus from upper respiratory samples but is still sick from COVID-19 and also for epidemiological purposes. With such a poor sensitivity of their assays it is likely that a large number of tests will end up with false negative results with adverse outcomes.

We thank the reviewer for this perspective. We completely agree that a diagnostic test is ideally sensitive enough to detect individuals with low viral loads. But even now, months after our initial submission, diagnostic qRT-PCR tests are still not widely available enough and the test presented in this manuscript can help fill in specific gaps in testing. 

RT-LAMP represents a low-cost, faster turn-around time, lower sensitivity test that can be used to screen individuals frequently who are otherwise asymptomatic and do not qualify for diagnostic testing at many sites that continue to have to limit the number of daily tests they run (2). Asymptomatic individuals have similar viral loads as symptomatic individuals and would be just as likely to test positive by this assay as symptomatic individuals. The use of the newly FDA emergency use-approved antigen tests, as another version of a lower sensitivity, fast turn-around test, has been promoted by the CDC as a way to augment testing (3). RT-LAMP has a similar capability to detect the individuals with high viral load who are likely the most contagious individuals. We have changed the emphasis of the importance of this test away from identifying highly contagious individuals and incorporated the importance of frequent screening and how this test can help in that capacity. This will be especially important as the massive fall/winter surge in the US declines after the arrival of vaccines and vigilance will be required to identify and suppress outbreaks from asymptomatic individuals throughout society.

These changes are presented in the introduction in lines 69-81. In addition, we have transitioned our figures and analysis to report the sensitivity of this assay as copies/ul rather than copies/ml. This transition is in line with most other RT-LAMP-based manuscripts and the way that the CDC reports the limit of detection of their PCR-based assay. This will hopefully avoid confusion when comparing these data with those of other published work. The sensitivity of our assay is not different from other RT-LAMP-based assays that do not use an RNA extraction first (~100 copies/ul).

2. The authors claim to apply their assay as POC is not very encouraging considering the fact that switching to colorimetric detection methods will further reduce the sensitivity of the test. 

Based on the recommendation of the reviewers, the authors are removing the colorimetric test results with saliva from this manuscript as it is outside of the scope of this work. To be clear, creating a test that can be simple enough to run and does not require expensive equipment, such as a lightcycler, are important steps to getting widespread SARS-CoV-2 screening available to schools and workplaces, who can run these tests on-site with some training. If we can screen workforces or schools and catch at least some of the most highly positive individuals, we hypothesize this will help reduce transmission risk, though we cannot predict by how much. More sensitive options are by nature too costly and time consuming to use in this capacity. This allows us to cast a much wider net with and assay that is comparable in sensitivity to antigen tests approved by the FDA, which also remain in limited supply and are not widely available. The advantages of this test, even if less sensitivity than qRT-PCR, are addressed in lines 69-81.

3. The specificity of any of the assays were not addressed. It is not clear what is the rate of false positive results from these assays.

The specificity of the test is determined by the primer sets used in the assay. Because the primer sets tested in this manuscript were previously published and many were shown to be specific for SARS-CoV-2 either in silico by sequence analysis or by testing against other coronaviruses in the lab, we did not reevaluate this. We have added a paragraph to the methods section about the specificity of the primer sets based on previous work by those who developed the primers in lines 137-146. We also added citations for each primer set in Table S1 for additional reference. We did test 5 samples from the hospital that were negative for SARS-CoV-2 by diagnostic tests but were positive for other respiratory pathogens that all tested negative by our LAMP assay using the gene-N-A primers. We included more details about these samples in lines 200-203. In addition, we tested 26 other SARS-CoV-2 negative samples from the hospital with the gene-N-A primers and all samples also tested negative by our LAMP test. Based on these results, specificity of the assay with the gene-N-A primers is 100% for SARS-CoV-2. We have added this to the text in lines 202-203. Unfortunately, we do not have access to a large number of other coronaviruses and primer sets to adequately address specificity. The primers described by Color genomics, which we recommend using based on the experimental results provided in our manuscript, were validated against 51 organisms and showed those primers were highly specific for SARS-CoV-2 as part of the EUA process for their assay that also uses these primers. This information was added in lines 145-146. 

4. No proper clinical validation was done. So, the clinical performance characteristics of the assays in comparison with standard RT-qPCR assays are not clear.

The authors acknowledge that we did not have the samples nor resources to provide a clinical validation of this particular assay. The authors are recommending this test as a non-diagnostic surveillance test, which does not require strict clinical validation. Instead, the authors wished to present their findings to provide their insight to the community at large who may be interested in developing this test in any number of capacities. If someone wishes to pursue this test in a clinical or diagnostic capacity, additional work would be required. This has been added to the discussion in lines 406-407.

5. It is somewhat surprising that the LOD for pure transcripts (Fig 1A) is two log higher than extracted RNA from samples (Fig 2B).

Thanks for pointing out this inconsistency. By pure transcript we can detect as low as 156 copies/ul (Figure 1) and with extracted RNA we can detect samples with a concentration as low as 10 copies/ul and consistently detect samples above 23 copies/ul. Please note that the RNA copies/ul reported is the concentration of the starting sample in Fig. 2B. During the process of RNA isolation, the RNA is purified and concentrated by 3-fold and then 5ul were used to emulate the volume and copies of RNA that goes into qRT-PCR for comparison. For a sample with 10 copies/ul, the total copies per reaction would be 10 copies RNA x 3-fold dilution x 5ul=150 copies/rxn. That is essentially the same number of copies/reaction detectable with the transcript where 1ul of a sample with 156 copies of RNA/ul were added. Overall, detection in transcript mimics that of detection in primary samples where RNA was isolated. This distinction between copies of RNA/ul in the starting sample and copies of RNA/reaction is important when interpreting the data shown in Figure 2B and for clarity we’ve added this description to the text in lines 257-264.

6. Different primer sets were tested in a much smaller set of samples compared to Gene-N-A primers.

It is true that we focused on the Gene-N-A primers for much of their work, including the work with primary samples. Due to limitations in acquiring primary samples and that they were used for multiple assays and were used up, we were not able to repeat some of our initial work with the alternative primers that we believe perform better than the Gene-N-A primers. Instead, we chose to share the work we did to determine whether different primer sets could improve the efficiency of the assay using standards and samples that we did have available.

7. Colorimetric saliva results were not correctly presented. The results need to be blindly called and then compared against NP swab RT-qPCR results.

Per request from the reviewers the colorimetric saliva results were removed from the manuscript.

8. There are now hundreds of RT-qPCR and RT-LAMP methods published including many extraction free PCR with much better analytical and clinical sensitivity. The authors need to discuss how their tests and evaluations provide additional benefits over existing assays.

This is a rapidly changing field and while at this point, there are other extraction-free PCR test options, limits of detection around 100 copies/ul are common with most of these methods. The primary contribution of this manuscript is that it shows results from performing the assay on primary NP swab samples. Most other publications utilize contrived samples with either virus or transcript that do not necessarily recapitulate the performance of the assay in primary samples containing biological inhibitors such as nucleases that will alter the performance (addressed in lines 64-68 and 341-348). While our LOD with transcript is around 156 copies/ul, in primary samples that falls closer to 1,000 copies/ul. 

As more assays evolve it is likely that modifications will improve this LOD and the authors believe it is important to present what we’ve learned to the community in order to aid in that development as quickly as possible. Lastly, as mentioned in response to question 1, the benefit of an assay like this that does not require extraction first is that is faster and cheaper than more sensitive molecular tests, which means it can be performed much more frequently and is likely to capture those that are most likely to transmit the virus despite being asymptomatic who are otherwise not getting tested. Different versions of assays may all have utility, as end-user preferences will likely factor into eventual acceptance for repeated testing. This emphasis has been incorporated into the introduction (lines 69-81). 

Footnotes:

1. https://www.medrxiv.org/content/10.1101/2020.07.28.20164038v2.

2.https://www.pewtrusts.org/en/research-and-analysis/blogs/stateline/2020/08/14/to-speed-up-results-states-limit-covid-19-testing;
https://www.businessinsider.com/why-states-like-new-york-are-limiting-covid-19-tests-2020-3;
https://www.wsj.com/articles/covid-19-testing-is-hampered-by-shortages-of-critical-ingredient-11600772400

3. https://www.cdc.gov/coronavirus/2019-ncov/lab/resources/antigen-tests-guidelines.html

---

## [Editor Report · Decision Letter 1]

18 Dec 2020

Optimizing direct RT-LAMP to detect transmissible SARS-CoV-2 from primary nasopharyngeal swab samples

PONE-D-20-29372R1

Dear Dr. Dudley,

We’re pleased to inform you that your manuscript has been judged scientifically suitable for publication and will be formally accepted for publication once it meets all outstanding technical requirements.

Kind regards,

Ruslan Kalendar, PhD

Academic Editor

PLOS ONE

---

## [Editor Report · Acceptance letter]

23 Dec 2020

PONE-D-20-29372R1 

Optimizing direct RT-LAMP to detect transmissible SARS-CoV-2 from primary nasopharyngeal swab samples 

Dear Dr. Dudley:

I'm pleased to inform you that your manuscript has been deemed suitable for publication in PLOS ONE. Congratulations! Your manuscript is now with our production department. 

Kind regards, 

on behalf of

Prof. Ruslan Kalendar 

Academic Editor

PLOS ONE